# PROBABILISTIC NEURAL TRANSFER FUNCTION ESTIMATION WITH BAYESIAN SYSTEM IDENTIFICATION

## ABSTRACT

Neural population responses in sensory systems are driven by external physical stimuli. This stimulus-response relationship is typically characterized by receptive fields, which have been estimated by *neural system identification* approaches. Such models usually requires a large amount of training data, yet, the recording time for animal experiments is limited, giving rise to epistemic uncertainty for the learned neural transfer functions. While deep neural network models have demonstrated excellent power on neural prediction, they usually do not provide the uncertainty of the resulting neural representations and derived statistics, such as the stimuli driving neurons optimally, from *in silico* experiments. Here, we present a Bayesian system identification approach to predict neural responses to visual stimuli, and explore whether explicitly modeling network weight variability can be beneficial for identifying neural response properties. To this end, we use variational inference to estimate the posterior distribution of each model weight given the training data. Tests with different neural datasets demonstrate that this method can achieve higher or comparable performance on neural prediction, with a much higher data efficiency compared to Monte Carlo dropout methods and traditional models using point estimates of the model parameters. At the same time, our variational method allows to estimate the uncertainty of stimulus-response function, which we have found to be negatively correlated with the predictive performance and may serve to evaluate models. Furthermore, our approach enables to identify response properties with credible intervals and perform statistical test for the learned neural features, which avoid the idiosyncrasy of a single model. Finally, *in silico* experiments show that our model generates stimuli driving neuronal activity significantly better than traditional models, particularly in the limited-data regime.

## 1 INTRODUCTION

Current neural interfaces allow to simultaneously record large populations of neural activity. In sensory neuroscience, such ensemble responses are driven by external physical stimuli (e.g., natural images), and their relation has been characterized by tuning curves or receptive fields (RFs; Hubel & Wiesel (1959)). Such stimulus-response functions have been estimated by *neural system identification* methods (reviewed in Wu et al., 2006). Classically, they used a linear-nonlinear-Poisson (LNP) model or variants of it (Chichilnisky, 2001; Pillow et al., 2008; Huang et al., 2021; Karamanlis & Gollisch, 2021) to predict responses to unseen stimuli such as white noise and natural images (Rust & Movshon, 2005; Qiu et al., 2021). More recently, deep neural networks (DNNs) with multiple layers of non-linear processing have shown great success for learning neural transfer functions along the ventral visual stages from retina (McIntosh et al., 2016; Batty et al., 2016; Qiu et al., 2023) and primary visual cortex (Antolík et al., 2016; Klindt et al., 2017; Ecker et al., 2018; Lurz et al., 2021) to higher visual areas (Yamins et al., 2014; Güçlü & van Gerven, 2015). Moreover, through *in silico* experiments, these models are able to generate specific stimulus to control neural activity and identify novel neuronal properties from a high-dimensional space (Bashivan et al., 2019; Ponce et al., 2019; Walker et al., 2019; Franke et al., 2021; Hoefling et al., 2022). For example, closed-loop paradigms show that performing gradient ascent on a deep model can yield most exciting inputs (MEIs) to drive a neuron's activity optimally (Walker et al., 2019).

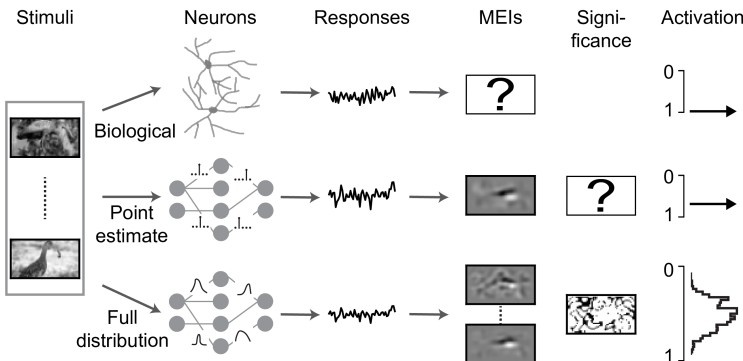

Figure 1: Schematic of neural system identification for predicting responses. Biological neurons (top row; second column) respond to visual stimuli (first column) distinctly (third column), with an unknown MEI (fourth column) driving a cell with optimal activation (sixth column). Traditional system identification methods (center row) learn stimulus-response function and yield a MEI with unknown statistics (fitth column). Bayesian approaches (bottom row) learn distributions of model parameters to predict neuronal responses, yielding an infinite MEIs, whose significance map can be computed by sampling from posterior, to drive a neuron with credible intervals.

Yet, these system identification approaches demand significant amounts of stimulus-response pair data for the model training, given the high dimensional stimulus space and the non-linear neural transformations (Lurz et al., 2021; Cotton et al., 2020; Qiu et al., 2023). Due to limited recording time for each experiment, the amount of data for fitting these models is restricted introducing epistemic uncertainty about the learned stimulus-response function. To estimate this uncertainty, traditional LNP methods obtain full posterior distribution of model parameters by leveraging a Bayesian framework to provide confidence intervals for the estimated RFs (Gerwinn et al., 2007; 2010; Park & Pillow, 2011; Huang et al., 2021). However, DNN models rarely consider the uncertainty of the neuronal properties that are recovered from *in silico* experiments.

Here, we propose a Bayesian system identification approach to estimate response features of neurons with uncertainties (Figure 1). We test whether incorporating uncertainties by learning the full distribution of model parameters is beneficial for learning neural representations. To this end, we build a DNN model to predict responses to unseen visual stimuli by using variational inference to estimate the distribution of network weights, i.e., Bayes by Backprop (Hinton & Van Camp, 1993; Neal & Hinton, 1998; Jaakkola & Jordan, 2000; Blundell et al., 2015).

Our contributions are: (1) We incorporate weight variability in deep neural networks for identifying neural response functions with uncertainty and extend the Bayes by Backprop with a hyperparameter which effectively adjusts sparsity of model parameters. (2) We apply our Bayesian models on different experimental datasets and find that our method can achieve higher or comparable performance on neural prediction, with a much better data efficiency, compared to Monte Carlo dropout methods and traditional models using point estimates of the model parameters. (3) Our Bayesian approach with full posterior allows to estimate neural features with credible intervals and run statistical test for the derived MEIs, bypassing the idiosyncrasy of a single model. (4) Finally, *in silico* experiments demonstrate that our variational model yields stimuli that drive neuronal activation better than the traditional models, especially in the condition of limited training data. This supports that weight uncertainty, as implemented in our model, may contribute to a more efficient identification of non-linear neuronal response functions.

## 2 METHODS

### 2.1 MODELS

**Variational model** DNN for system identification can be seen as a probabilistic model: given the training data $\mathcal{D} = (\mathbf{x}_i, \mathbf{y}_i)_i$ where $\mathbf{x}_i$ is an input (such as natural images) and $\mathbf{y}_i$ is the output

(such as neural responses), we aim to learn the weights $\mathbf{w}$ of a network which can predict the output for the unseen stimuli (Figure 1). Compared to a traditional method using point estimates of the weights, Bayesian approaches learn full distributions of these $\mathbf{w}$. Estimating the full posterior distribution of the weights $P(\mathbf{w}|\mathcal{D})$ given the training data is usually not feasible. An alternative is to approximate $P(\mathbf{w}|\mathcal{D})$ by a new distribution $q(\mathbf{w}|\theta)$ whose parameters $\theta$ are trained to minimize the distance between the proxy and the true posterior, which is called variational inference (Hinton & Van Camp, 1993; Neal & Hinton, 1998; Jaakkola & Jordan, 2000; Blundell et al., 2015). Usually we use Kullback-Leibler (KL) divergence as a measure of distance between two distributions:

$$\theta^* = \arg\min_\theta \mathbf{KL}[q(\mathbf{w}|\theta)||P(\mathbf{w}|\mathcal{D})] \tag{1}$$

$$= \arg\min_\theta \mathbf{KL}[q(\mathbf{w}|\theta)||P(\mathbf{w})] - \mathbb{E}_{q(\mathbf{w}|\theta)}[\log P(\mathcal{D}|\mathbf{w})] \tag{2}$$

The optimization function can be viewed as a trade-off between the distance between the variational posterior and the selected prior and the likelihood cost. We can view it as a constrained optimization problem as (Higgins et al., 2016):

$$\arg\max_\theta \mathbb{E}_{q(\mathbf{w}|\theta)}[\log P(\mathcal{D}|\mathbf{w})] \quad \text{subject to} \quad \mathbf{KL}[q(\mathbf{w}|\theta)||P(\mathbf{w})] < \epsilon \tag{3}$$

Here $\epsilon$ represents the specific distance between the variational posterior and the prior. According to KKT conditions (Kuhn & Tucker, 1951) and non-negative properties of KL divergence, we get:

$$\mathcal{F} = \mathbb{E}_{q(\mathbf{w}|\theta)}[\log P(\mathcal{D}|\mathbf{w})] - \beta_v(\mathbf{KL}[q(\mathbf{w}|\theta)||P(\mathbf{w})] - \epsilon) \tag{4}$$

$$\geq \mathbb{E}_{q(\mathbf{w}|\theta)}[\log P(\mathcal{D}|\mathbf{w})] - \beta_v\mathbf{KL}[q(\mathbf{w}|\theta)||P(\mathbf{w})] \tag{5}$$

where $\beta_v$ is non-negative and represents a Lagrangian multiplier. So the final loss function for the model is:

$$\mathcal{L} = \beta_v\mathbf{KL}[q(\mathbf{w}|\theta)||P(\mathbf{w})] - \mathbb{E}_{q(\mathbf{w}|\theta)}[\log P(\mathcal{D}|\mathbf{w})] \tag{6}$$

$$\approx \sum_{i=1}^n \beta_v(\log q(\mathbf{w}^{(i)}|\theta) - \log P(\mathbf{w}^{(i)})) - \log P(\mathcal{D}|\mathbf{w}^{(i)}) \tag{7}$$

Eq. (7) is a result of Monte Carlo sampling $n$ instances $\mathbf{w}^{(i)}$ from $q(\mathbf{w}|\theta)$ because we can not calculate (6) directly.

Here, we implemented convolutional neural networks (CNNs) for all experiments. For a CNN using variational inference on model weights (variational model), we picked independent Gaussian distributions for the variational posterior and a scale mixture of two Gaussians for the prior (Blundell et al., 2015). The log posterior was defined as $\log q(\mathbf{w}|\theta) = \sum_{k=1} \log \mathcal{N}(w_k|\mu, \sigma^2)$ where $w_k$ denotes $k$th weight of the neural network and $(\mu, \sigma)$ are the posterior parameters $\theta$. To keep $\sigma$ non-negative, we parameterised it using $\sigma = \log(1 + \exp(\rho))$. We selected the log prior $\log P(\mathbf{w}) = \sum_{k=1} \log(\pi\mathcal{N}(w_k|0, \sigma_1^2) + (1-\pi)\mathcal{N}(w_k|0, \sigma_2^2))$ where $\pi$ is a mixture component weight ($0 \leq \pi \leq 1$) (Blundell et al., 2015; Fortuin et al., 2021). This prior, compared to a single Gaussian distribution, encourages sparseness in learned kernels, reminiscent of neural representations in visual systems (Field, 1994; Olshausen & Field, 1996; David et al., 2007; Stevenson et al., 2008). The likelihood loss depends on the specific task of the network. For neural system identification, we use Poisson loss $-\log P(\mathcal{D}|\mathbf{w}) = \sum_l \hat{\mathbf{r}}_l - \mathbf{r}_l \log \hat{\mathbf{r}}_l$, where $l$, $\hat{\mathbf{r}}_l$ and $\mathbf{r}_l$ denote neuronal index, prediction responses and true responses, respectively.

**Baseline and control models** We used a CNN without any regularization as a baseline model (Appendix A.1) and used a CNN with L2 regularization in each convolutional layer and L1 regularization in fully connected layer (L2+L1) as a control model. We adopted an ensemble of L2+L1 models with different initialization seeds as a second control model, whose predicted responses are the average of five model outputs. To examine the contribution from weight uncertainties, we built a maximum a posteriori (MAP) model which contains prior and likelihood terms in Eq. (7) as loss functions. Additionally, as a fourth control, we adopted a CNN with Monte Carlo dropout for probabilistic prediction; it used the same dropout rate for each model layer and in both training and test stages (Srivastava et al., 2014; Gal & Ghahramani, 2016).

## 2.2 DATASET

We tested our method on two publicly available datasets.

The first dataset contains calcium signals driven by static natural gray-scale images for neurons in primary visual cortex (V1) of mice (Antolík et al., 2016). We used 103 neurons from the first scan field, whose single-trial responses to 1,600 images for training models and 200 for tuning hyperparameters. Then we used the mean of response repeats to 50 test images for evaluating models.

The second dataset comprises $Ca^{2+}$ responses to natural green/UV images (36x64 pixels) for neurons in mouse V1 (Franke et al., 2021). We selected the natural stimuli that were presented in both UV and green channels and used the neurons whose quality index ($QI = \text{Var}[\text{E}[C]_r]_t / \text{E}[\text{Var}[C]_t]_r$, time samples $t$ and repetitions $r$, a response matrix $C$ with a shape of $t \times r$, $\text{E}[X]_d$ and $\text{Var}[X]_d$ denoting the mean and variance along the dimension $d$ of $X$, respectively) of 10-repeat test responses were larger than 0.3. In this way, we had 161 neurons from one scan field, whose single-trial responses to 4,000 images for training and 400 for validation. Then we used mean of response repeats to 79 test images for evaluation.

## 2.3 TRAINING AND EVALUATION

We trained all models with a learning rate of 0.0003 for a maximum of 200 epochs using the Adam optimizer (Kingma & Welling, 2013). We computed linear correlation between predicted and recorded responses, which was used to evaluate models on validation or test data. We tuned model hyperparameters and selected the ones as well as the respective epoch number with the best predictive performance on validation data. To keep the comparison fair, the test models shared similar network architecture for each dataset, except that the dropout model featured dropout layers.

For each trained model, we estimated MEIs of all neurons by running gradient ascent on a random input image for 100 steps with a learning rate of 10 and we picked the stimulus with the highest activity (Erhan et al., 2009; Walker et al., 2019). All generated MEIs had the same mean and standard deviations as the training images. For the two probabilistic (variational and dropout) models, we ran the estimation for 100 times with Monte Carlo sampling, hence, we got 100 MEIs (matrix $C$) for each recorded neuron. We defined MEI variance of one neuron as $\text{MEI variance} = \text{E}[\text{Var}[C]_s]_{hw}$ (sampling times $s$, stimulus height $h$, stimulus width $w$, and $C$ with a shape of $s \times h \times w$). The overall MEI variance for a model was an average of MEI variances for the recorded neurons.

In *in silico* experiments, to measure the activation distribution of MEIs yielded from variational models for a neuron, we estimated 100 MEIs by sampling and one mean MEI by using the weight mean $\mu$ from each seed. So we had 505 MEIs for five random seeds with one additional MEI which was the mean of the five mean MEIs, in total 506 MEIs. For L2+L1 models, we estimated five MEIs from different random seeds and also got one by averaging across these MEIs, in total 6 MEIs.

## 3 RESULTS

### 3.1 $\beta_v$ BALANCES MODEL CAPACITY AND DATA LIKELIHOOD

We first analyzed the possible roles of $\beta_v$ in the loss function of variatonal models. Eq. (7) has a similar form with the objective functions in deep variational information bottleneck (Alemi et al., 2016; Tishby et al., 2000) and $\beta$-VAE (Higgins et al., 2016; Burgess et al., 2018), inspiring us to investigate it from the perspective of information theory.

The training objective jointly minimizes the KL divergence between the posterior $q(\mathbf{w}|\theta)$ and the prior $P(\mathbf{w})$ and maximizes the data likelihood under the distribution $q(\mathbf{w}|\theta)$. The distribution distance becomes zero when $q(\mathbf{w}|\theta) = P(\mathbf{w})$. In the case of Gaussian posterior and heavy-tailed prior with mean zero, the divergence decreases with the posterior mean moving close to zero and the posterior variance decreasing, which induces many zeros for weights $\mathbf{w}$ and increases the sparsity of model parameters. In the extreme case, all weights are equal to zeros and the model does not have any expressive power. In such case, the log likelihood $\mathbb{E}_{q(\mathbf{w}|\theta)}[\log P(\mathcal{D}|\mathbf{w})]$ vanishes, indicat-

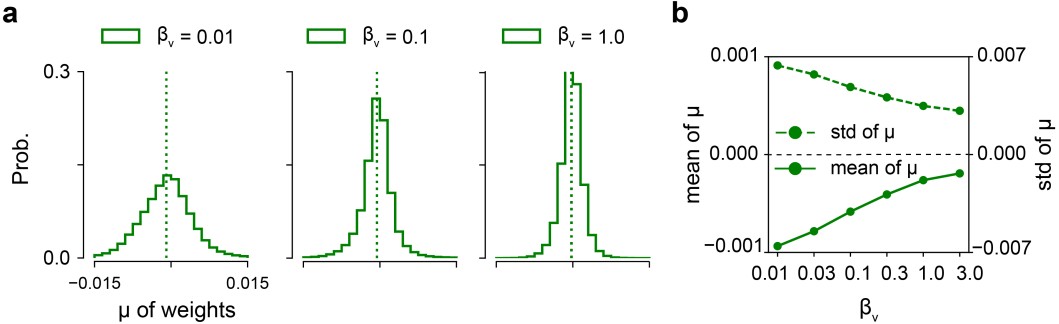

Figure 2: Hyperparameter $\beta_v$ for regulating weight sparseness. **(a)** Distribution of the means ($\mu$) of model weights for different $\beta_v$ values. Dotted lines indicate distribution means. **(b)** Mean and standard deviation for the distributions in (a).

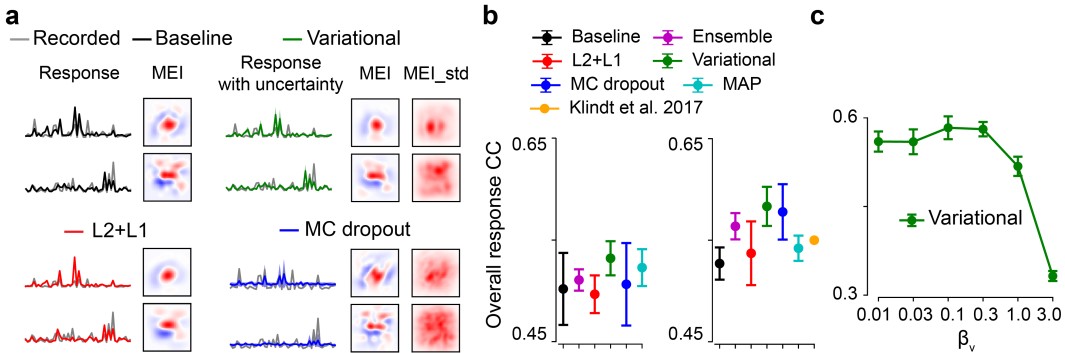

Figure 3: Neural prediction with weight uncertainty. **(a)** Mean recorded responses (gray) and predictive responses to natural stimuli(black, baseline; red, L2+L1; green, variational one with $\beta_v = 0.1$; blue, MC dropout with dropout rate 70%; shaded green and blue representing standard deviation for the variational and the dropout methods, respectively), estimated MEIs, as well as standard deviation of MEI (MEI_std; only for two probabilistic models), for two exemplary neurons. MEI and MEI_std use different color scales with red and blue indicating positive and negative values, respectively. **Note that MEI has much larger absolute values than MEI_std.** **(b)** Predictive performance (correlation coefficient, CC) based on test data with different amounts of training data (left, 50% of training data; right, 100% of data) for 6 models (purple, ensemble; cyan, MAP; 10 seeds per model), and the one used by (Klindt et al., 2017) (orange). **(c)** Predictive model performance for different $\beta_v$ values. Error bars in (b) and (c) represent standard deviation of n=10 random seeds for each model.

ing that the posterior $q(\mathbf{w}|\theta)$ is a bottleneck for maximizing the data likelihood. Therefore, $\beta_v$ can be interpreted as a coefficient to adjust model expressive power for fitting the data.

Empirically, we examined the distribution of weight means ($\mu$) for different $\beta_v$ values on the dataset 1 shared by by Antolik and colleagues (Antolík et al., 2016). Indeed, we found that with the increase of $\beta_v$, the mean of the distribution got close to zeros and the std decreased, indicating an increase of sparsity of model weights (Fig. 2). Therefore, the hyperparameter $\beta_v$ served to tune the model capacity via weight sparseness for data prediction.

### 3.2 SYSTEM IDENTIFICATION INCORPORATES MODEL UNCERTAINTY TO PREDICT NEURAL RESPONSES

We trained the six models on the dataset 1 (Fig. 3a) and tuned their respective hyperparameters using validation data. For the variational model, we found the one with $\beta_v = 0.1$ had best predictive performance with a sharp decrease when increasing $\beta_v$ till 1.0 or 3.0 (Appendix A.2.1). We also

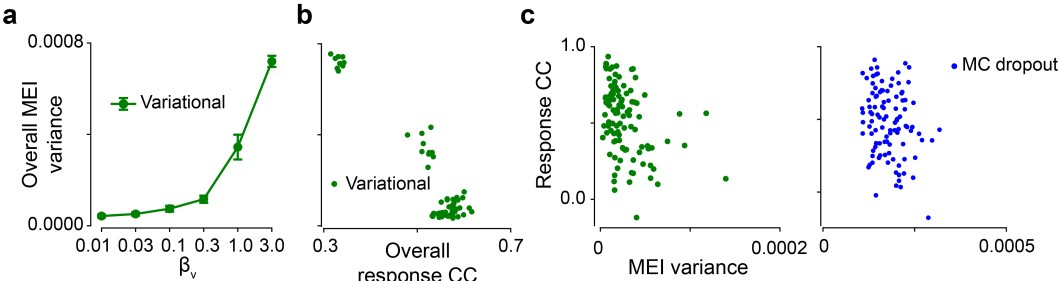

Figure 4: Neural transfer functions with variability. **(a)** Overall MEI variance for different $\beta_v$ values (10 seeds per model). **(b)** Scatter plot of overall response CC and overall MEI variance for 6 $\beta_v$ values and 10 seeds (each dot representing one model at each $\beta_v$ and each seed). **(c)** Scatter plot of response CC and MEI variance for two probabilistic models at one random seed (each dot representing one neuron). Error bars in (a) represent standard deviation of n=10 random seeds for each model.

observed that at training stage, the variational model presented a more stable performance on validation data compared to the baseline CNN, confirming the regularization effect of prior to prevent overfitting.

Next, we selected the hyperparameters achieving the best performance on validation data for each model. To examine the feature properties learned by these models, we estimated the MEIs of recorded neurons and found that these models yielded antagonistic center-surround and Gabor filters in a local region, reminiscent of neural representations in early visual processing ((Hubel & Wiesel, 1959; Chichilnisky, 2001); Fig. 3a). To compare the performance of neural prediction, we then evaluated all models using test data. Interestingly, when we used the full training data, the variational and MC dropout models had similar predictive performance with a slightly higher value for the variation one ($p = 0.6159$, two-sided permutation test with n = 10,000 repeats). The variational one also outperformed the baseline, the L2+L1, the ensemble, the MAP ($p = 0.0001$) and the model with shared feature space between neurons((Klindt et al., 2017); Fig. 3b). With half of training data, the variational method yielded a correlation higher compared to the MC dropout method ($p = 0.082$), but slightly better than the MAP one ($p = 0.2526$). The performance difference between variational and traditional methods using point estimates of parameters indicates the benefit of weight uncertainty for neural prediction. We then reanalyzed the influence of $\beta_v$ on prediction for the variational model using test data. Similar to the case with validation data, we noticed a rather steady predictive performance with increasing $\beta_v$ until a sudden drop at $\beta_v = 1.0$ or 3.0, implying that a large Lagrangian multiplier imposing excessive sparsity on weights yields model underfitting.

Together, the superior/equivalent performance of our variational approach suggests that incorporating weight uncertainty is beneficial for predicting neural responses.

### 3.3 PROBABILISTIC MODELS LEARN VARIANCE OF NEURAL TRANSFER FUNCTIONS

The variational and the MC dropout approaches enable us to learn stimulus-response functions with credible intervals. We next asked whether the variability of the learned transfer function was related to the predictive performance for the two probabilistic (variational and dropout) models. To this end, we measured the MEI variance for each neuron and the overall MEI variance for each model and relate them to the performance on predicting responses.

We first investigated the influence of $\beta_v$ on the variability of the learned transfer functions for our variational model. Interestingly, we found a sudden increase of overall MEI variance at $\beta_v = 1.0$ or 3.0 (Fig. 4a), where an abrupt drop of model performance was present (cf. Fig. 3c). This opposite change between MEI variability and predictive performance was confirmed by the negative correlations between overall MEI variance and overall response CC ($r = -0.95, p < 0.0001$; Fig. 4b). Additionally, this negative correlation was also reflected at neuronal level. Both the variational and the MC dropout models had a negative correlation between response CC and MEI variance for the recorded neurons ($r = -0.37, p = 0.0001$ and $r = -0.23, p = 0.02$ for the variational and the

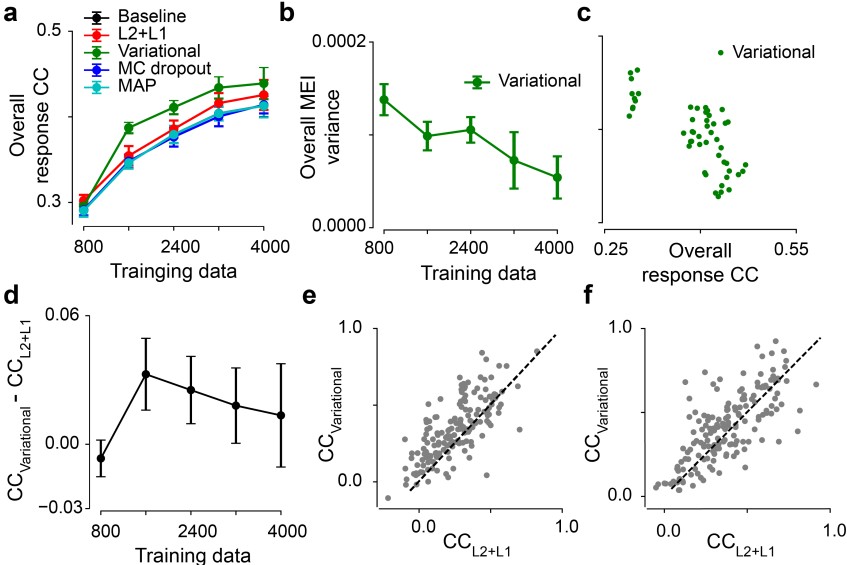

Figure 5: Variational models on the second dataset. **(a)** Model performance based on test data of the second dataset with different amounts of training data for five models (n=10 random seeds per model). **(b)** Overall MEI variance for different amounts of training data for variational models (10 seeds per model). **(c)** Scatter plot for overall response CC and overall MEI variance for different amounts of training data and at 10 seeds. Each dot representing one model. **(d)** Performance difference between the variational and the L2+L1 models. **(e)** Scatter plot of model predictions for the variational model and the L2+L1 model at one random seed when using 40% training data. Each dot representing one neuron. **(f)** Like (e) but using 100% training data. Error bars in (a), (b) and (d) represent standard deviation of n=10 random seeds for each model.

MC dropout, respectively; Fig. 4c), indicating that, for a trained probabilistic model, neurons with higher predictive performance have higher confidence on its estimated MEI.

In summary, these results demonstrate that a probabilistic model with smaller uncertainty on the learned stimulus-response function yields higher predictive performance.

### 3.4 VARIATIONAL MODEL FEATURES HIGH DATA EFFICIENCY ON NEURAL PREDICTION

Here we applied our method on the second dataset shared by Franke and colleagues ((Franke et al., 2021)). After hyperparameter tuning, we selected $\beta_v = 0.3$ for the variational network and evaluated the five models on test data.

We first examined the relationship between the uncertainty of the learned stimulus-response function and the performance on predicting responses. We expect that, with more data used for training, the model yields better prediction along with smaller variance for the learned MEIs. We focused on the variational method. Indeed, when more training data was used, the predictive model performance increased (Fig. 5a) while the overall MEI variance decreased Fig. 5b, with a negative correlation between them ($r = -0.73, p < 0.0001$; Fig. 5c). Note that we did not observe a steady decrease of the overall response variance (Appendix A.2.2).

Next, we investigated whether the performance difference between the variation and the L2+L1 model was sensitive to the training data size (Fig. 5d). We observed that the variational method had higher correlations except for the case of extremely little data (20%). The difference peaked at 40% with an increase of 9% ($p < 0.0001$, two-sided permutation test with n = 10,000 repeats) and gradually decreased with more training data, indicating the benefit of variational inference for system identification. We also compared the predictive performance on individual neurons at one random seed, the Bayesian model outperformed the L2+L1 one for the conditions of 40% ($p < 0.0001$) and 100% ($p = 0.0927$) training data.

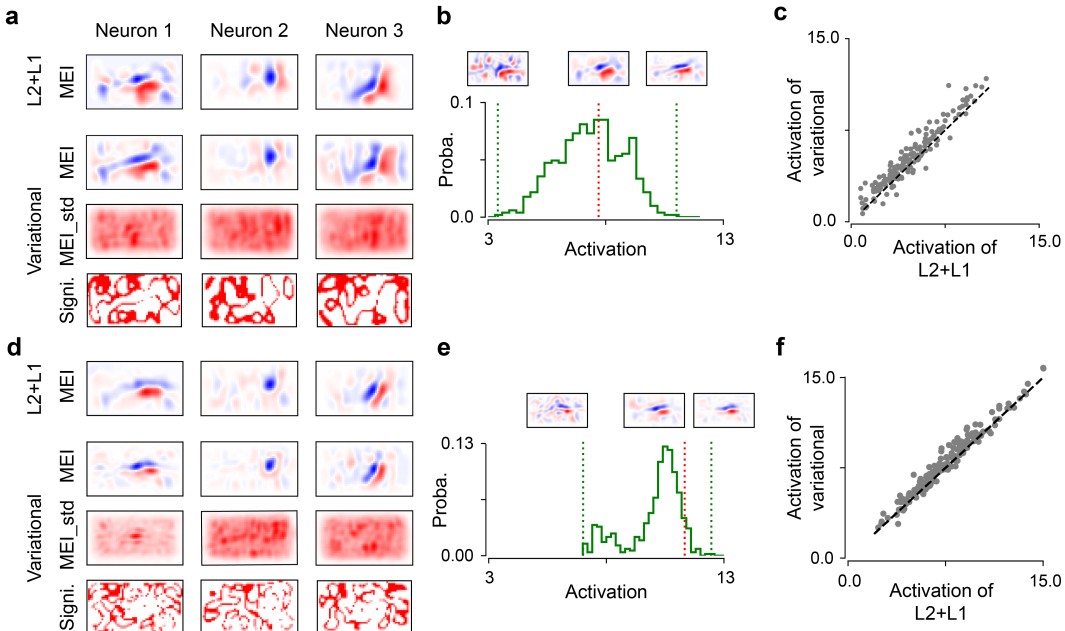

Figure 6: *In silico* experiments of neuronal activity with derived MEIs. **(a)** Estimated MEIs for L2+L1 (first row) and variational (second row) models, MEI_std (third row), as well as significance map (fourth row; white, $p < 0.01$, one-sample two-sided permutation test against zero for 10,000 repeats), for three exemplary neurons when using 40% of training data. MEI and MEI_std in the UV channel, with different color scales. **Note that MEI has much larger absolute values than MEI_std**. **(b)** 1D histogram of neuronal activity driven by the generated MEIs from the variational model for Neuron 1 when using 40% of training data. Insets: example MEIs with corresponding activation indicated by dotted lines (red, maximum of L2+L1; green, variational). **(c)** Scatter plot of activation driven by MEIs yielded from variational (using the weight mean $\mu$) and L2+L1 models at one random seed when using 40% of training data. Each dot representing one cell. **(d,e,f)** Same with (a), (b) and (c), but using 100% of training data.

Together, compared to a traditional method, our Bayesian approach with weight uncertainty yielded higher predictive performance with a higher data efficiency.

### 3.5 VARIATIONAL MODEL YIELDS STIMULI DRIVING HIGH NEURONAL ACTIVATION

Bayesian methods with full posterior provide an infinite ensemble of models for computing MEIs and allow to perform statistical tests for the derived features. We focused on the model using variational inference and the one using L2 and L1 regularization with 40% and 100% of training data. We found that these learned filters resembled neural features in the early visual system (Hubel & Wiesel, 1959; Chichilnisky, 2001) and localized more in the visual field with more training data (Fig. 6a,d). Like for the first dataset (Fig. 3a), MEI_std was not uniform across visual space, e.g., some presented Gaussian or bar shapes. Additionally, we examined whether the posterior of each pixel differs significantly from zero for the 100 sampled MEIs and found that the significance map may indicate zero-crossings in visual representations.

To further assess which method generates the more exciting stimuli for each cell, we conducted *in silico* experiments using a held-out L2+L1 model trained by full data as a digital testbed. For an example neuron, we measured the responses for all the 506 MEIs yielded from five variational models, and observed that these stimuli drove this neuron with quite different activity, with the maximum response larger than the maximum one yielded (from 6 MEIs) by the traditional models (Fig. 6b,e). With more training data, the activation distribution shifted towards higher mean with smaller variance. Additionally, we compared the activation on individual neurons for two methods (Fig. 6c,f), and observed that the Bayesian approach yielded higher responses for both conditions

using 40% ($p = 0.0473$, two-sided permutation test with n = 10,000 repeats) and 100% ($p = 0.2114$) of training data.

In summary, our variational model allowed statistical test for the derived response functions and yielded the stimuli driving neurons better than traditional methods, suggesting that weight uncertainty benefits the learning of neural representations.

## 4 DISCUSSION

We presented a Bayesian approach for identification of neural properties by incorporating model uncertainty through learning the distribution of model weights, aiming to estimate neural features with credible intervals. Our empirical results on different datasets show that the variational method had higher or comparable predictive performance, especially in the limited data regime, compared to methods using dropout or traditional methods learning point estimates of model parameters. Moreover, by sampling from posterior distribution of model weights, our approach enabled to provide credible intervals and test statistics for the learned MEIs, avoiding the idiosyncrasy of a single model. Finally, *in silico* experiments show that the variational model yielded the MEIs driving neurons with higher activity compared to the traditional model, in particular when limited data were used for training. This suggests that model uncertainty contributes to learning neural transfer functions with a high data efficiency.

**Relation to noise correlation**   Neural information process is probabilistic, i.e., neurons respond with trial-to-trial fluctuations to a repeated presentation of a stimulus (Perkel et al., 1967; Stein, 1967). Response variability is found across neural systems, originating from diverse factors, such as synapse variation, channel noise, brain state, and attention (Faisal et al., 2008; Mitchell et al., 2009; Cohen & Newsome, 2008; Cohen & Maunsell, 2009; Ecker et al., 2010; 2014). Additionally, the variability between populations of neurons are correlated. In a simplified case, a pair of neurons may present correlations for the single-trial responses, i.e., pairwise noise correlation, which also contributes to neural coding ((Abbott & Dayan, 1999); reviewed in (Averbeck et al., 2006; Kohn et al., 2016; Doiron et al., 2016; Da Silveira & Rieke, 2021)). Such response variability is inherent in neural data itself and is a kind of aleatoric, but not the epistemic uncertainty. We note that the standard deviations of the estimated MEIs from our models decreased with the increasing amounts of training data, suggesting that the variability of the sampled predicted responses may not be related to the response uncertainty in biological neurons or our models may predict a mix of both uncertainties (Appendix A.2.3).

**Future work & general impact**   While capturing the mean response, our variational approach incorporating model uncertainty did not predict the trial-to-trial variability. Such response fluctuation depends on many conditions, including biochemical process, internal brain states and engaged behavioral tasks (Faisal et al., 2008; Mitchell et al., 2009; Ecker et al., 2014; Goris et al., 2014). These factors have been described by a low-dimensional latent state models (Yu et al., 2008; Ecker et al., 2014; Bashiri et al., 2021). Therefore, a potential extension of our method could be a variational network incorporated with latent state variables.

Our *in silico* experiments indicate that the stimuli generated by the variation model driving higher neuronal activation than the CNN with regularization, which requires future animal experiments to test. Additionally, we noticed that the MEI_std was not uniform in the visual field for each neuron and its location was not overlaid with the central MEI, for example, it seems to sit on the surround of the corresponding MEI. It would be interesting to examine and quantify the MEI uncertainty in regard of visual space, which might be related to contextual sensory processing (Hock et al., 1974; Chiao & Masland, 2003; Fu et al., 2023).

More generally, why do we care about the uncertainty of the estimated neural representations? Even with closed-loop experiments, it is impossible for us to test all potential (exciting) inputs for the recorded neurons (Walker et al., 2019; Franke et al., 2021). Therefore, we always expect to have a confidence interval for the test statistics. Besides, a Bayesian model offers a manner to generate many stimulus candidates by sampling for stimulating neural systems, which may offer new insights for understanding the biological computation.

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
