# A  APPENDIX

## A.1  MODEL DETAILS

The CNN model for the first dataset shared by Antolik and colleagues consisted of a convolutional layer (24x1x9x9, output channels x input channels x image width x image height), a rectified linear unit (ReLU) function, another convolutional layer (48x24x7x7, output channels x input channels x image width x image height), another ReLU function, and — after flattening all dimensions — one fully connected (FC) layer (103x13872, output channels x input channels), followed by an exponential function. We used stride=1 and no padding for both convolutional layers. We trained the four models and tuned their respective hyperparameters. For the variational one, we tested different parameters for prior distribution on validation data, such as $\pi = 0$ or $\pi = 0.5$, $\sigma_1 = 1$ or $\sigma_1 = 100$, $\sigma_2 = \exp{(-3)}$ or $\sigma_2 = \exp{(-6)}$, and found that a scale mixture of two Gaussians had similar predictive performance, higher than one Gaussian distribution. As the predictive performance was similar for distinct priors on model layers, we used the same prior distribution with parameters $\pi = 0.5, \sigma_1 = 1, \sigma_2 = \exp{(-6)}$ for all layers. We also examined the number of Monte Carlo sampling times for model training and found that the predictive performance was similar for different numbers. Therefore, we used 1 or 2 sampling times for all model training.

The CNN model for the second dataset shared by Franke and colleagus contained a convolutional layer (48x2x9x9), a ReLU function, another convolutional layer (48x48x7x7), another ReLU function, and one FC layer (161x52800), followed by an exponential function. We used stride=1 and no padding for both convolutional layers. We used stride=1 and no padding for both convolutional layers.

For each CNN model, we tested different numbers (1-5) of nonlinear functions and different channel numbers (8, 16, 24, 32, 40 and 48) of each convolutional layer, and selected the ones which yielded (near) optimal predictive performance on validation data. We used small numbers when the performance was similar across models. For each dataset, the six methods used similar model architecture with the baseline CNN, except that the dropout model had dropout layers after the two ReLU functions and the FC layer.

We ran the neural prediction for 100 sampling times to get the uncertainties of responses to test stimuli. We defined variance of predicted response for one neuron as $\mathrm{Response\ variance} = \mathrm{E}[\mathrm{Var}[D]_s]_n$ (sampling times $s$, test stimulus number $n$, and response matrix $D$ with a shape of $s \times n$). The overall response variance for a model was an average of response variances for the recorded neurons. To calculate the variance of recorded response for a neuron, we replaced the sampling times with the repeated times of the presented test stimulus.

## A.2  ADDITIONAL RESULTS

### A.2.1  NEURAL PREDICTION FOR FIRST DATASET

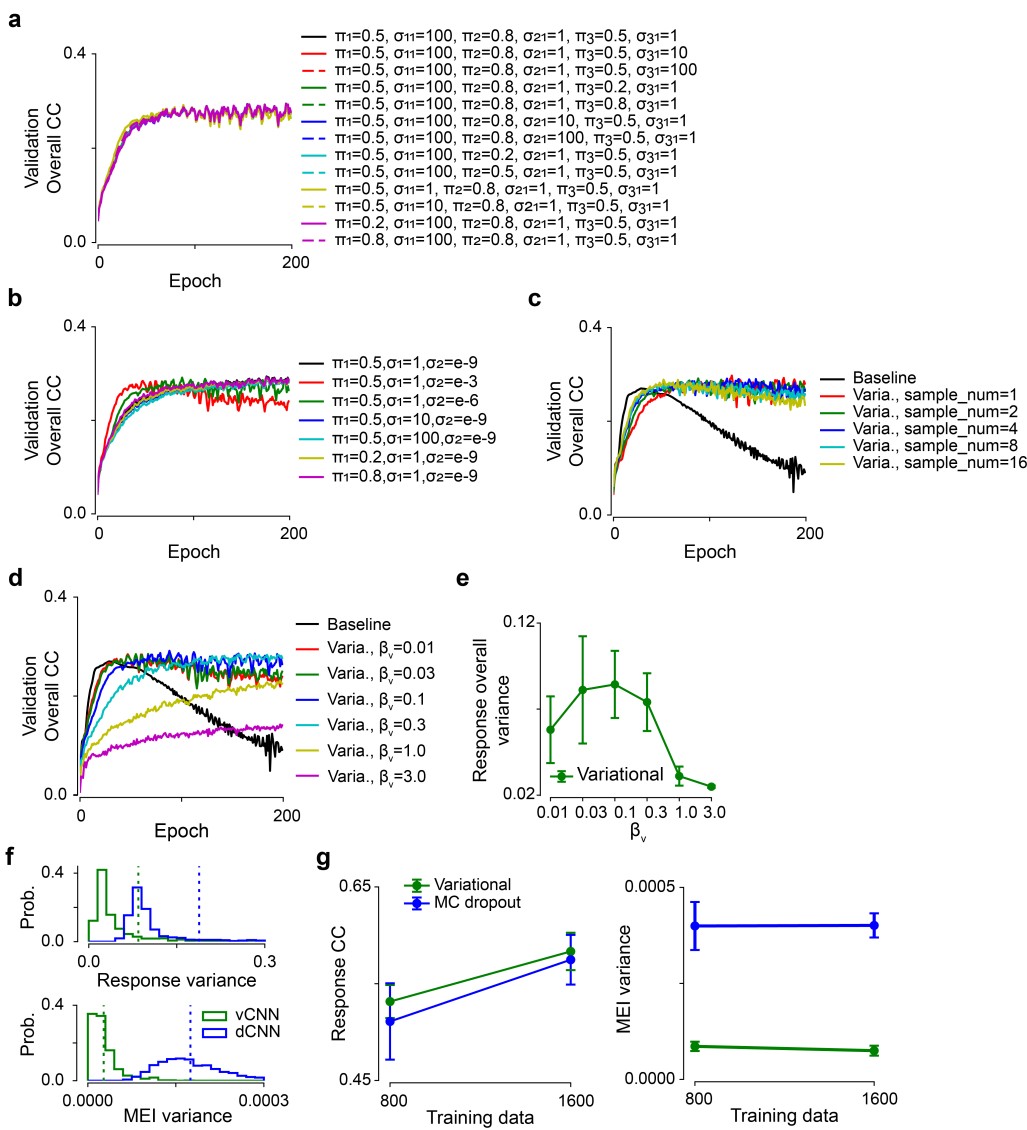

Figure 7: Neural prediction for first dataset. **(a,b)** Predictive performance (correlation coefficient, CC) based on validation data during training for variational models ($\beta_v = 0.1$) with different prior distributions. All layers adopted the same $\sigma_2 = \exp(-6)$ with different $\pi$ and $\sigma_1$ values (a), or with the same parameters of prior distribution (b). We picked $\pi = 0.5, \sigma_1 = 1, \sigma_2 = \exp(-6)$ for subsequent model training. **(c)** Predictive performance based on validation data during model training for different numbers of Monte Carlo sampling. We picked number=1 or 2 to save training time. **(d)** Model performance based on validation data during training for the baseline and variatonal models with different $\beta_v$ values. **(e)** Overall variance of predicted responses to test stimuli for different $\beta_v$ values. **(f)** Histogram of response variance (top) and MEI (RF) variance (bottom) for the variational and the MC dropout models. Dotted line represents the mean of histogram. **(g)** Model performance (left) based on test data and RF overall variance (right) for two probabilistic models with different amounts of training data. Error bars in (e) and (g) represent standard deviation of n=10 random seeds for each model.

### A.2.2 NEURAL PREDICTION FOR SECOND DATASET

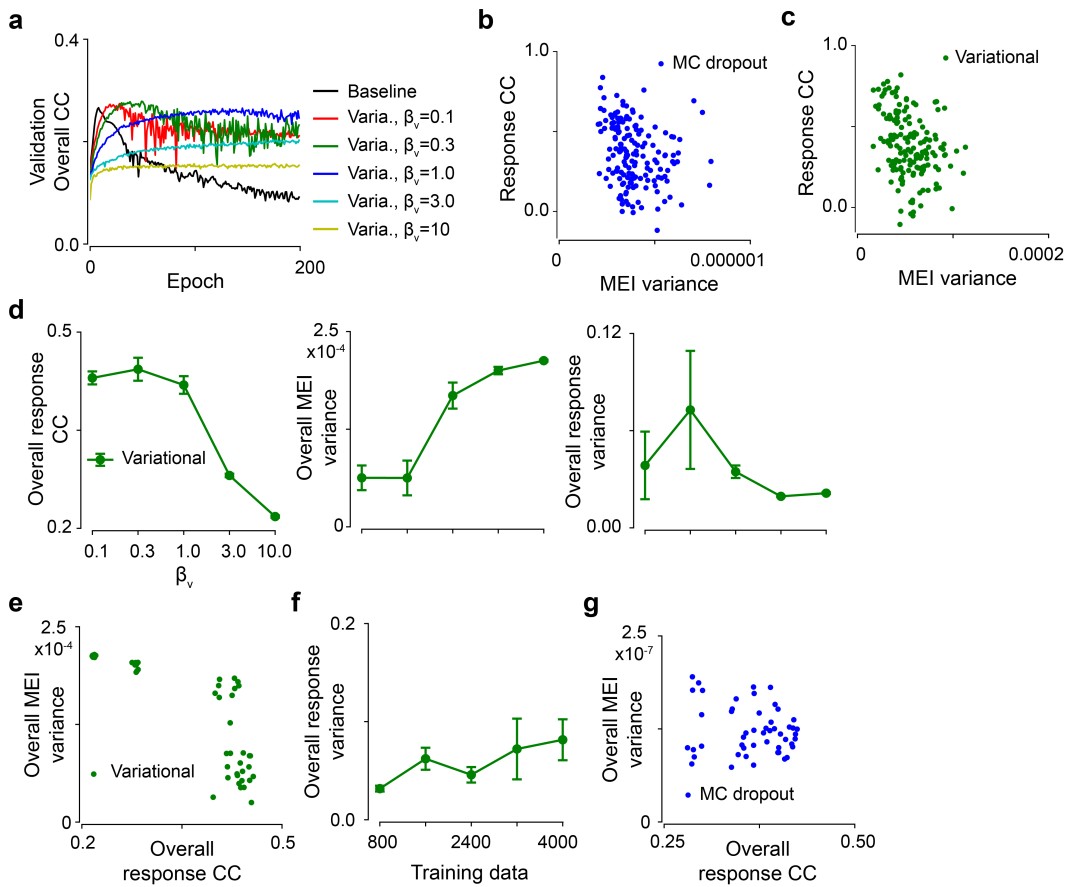

Figure 8: Neural prediction for second dataset. **(a)** Model performance based on validation data during training for the baseline and the variational models with different $\beta_v$ values. **(b,c)** Scatter plot of response CC and MEI (RF) variance for MC dropout (b) and variational (c) models for 10 seeds ($CC = -0.25, p = 0.001$ and $CC = -0.34, p < 0.0001$ for dropout and variational one, each dot representing one neuron at one random seed). **(d)** Predictive performance, overall RF variance and overall response variance for variational models with different $\beta_v$ values. **(c)** Predictive performance based on validation data during model training for different numbers of Monte Carlo sampling. We picked number=1 or 2 to save training time. **(d)** Model performance based on validation data during training for the baseline and the variational ones with different $\beta_v$ values. **(e)** Scatter plot for overall response CC and overall RF variance for the variational methods with different $\beta_v$ values (d) and at 10 seeds ($CC = -0.82, p < 0.0001$). Each dot represents one model. **(f)** Overall response variance for different amounts of training data for the variational models (10 seeds per model). **(g)** Scatter plot for overall response CC and overall RF variance for the dropout model with different amounts of training data and at 10 seeds ($CC = -0.17, p = 0.24$). Each dot represents one model. Error bars in (d) and (f) represent standard deviation of n=10 random seeds for each model.

### A.2.3 VARIANCE OF PREDICTED VS. RECORDED RESPONSES FOR SECOND DATASET

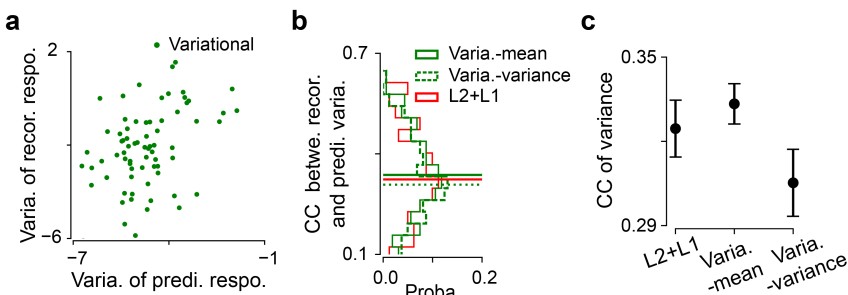

Figure 9: Variance of predicted vs. recorded responses for second dataset. Using the trained models, we tested whether the variance of predicted responses was related to the variance of recorded responses for each neuron. We first estimated the predicted response variance to a stimulus. For the L2+L1 model, as the mean of neural responses is proportional to the variance, we used the model output (a single predicted value) as a substitute. For the variational one, we either used the mean of predicted responses (multiple sampling times) as a substitute or calculated the response variance explicitly. **(a)** Scatter plot (axes in log scale) of predicted response variance (using response mean as a substitute) and recorded response variance for one neuron for a variational model. Each dot representing one stimulus. **(b)** Distribution of correlations between recorded and predicted response variance for all neurons for the L2+L1, variational-mean (using response mean as a substitute) and variational-variance (calculating response variance), at one random seed. Horizontal lines representing distribution means. **(c)** Mean correlations between two response variances (10 seeds per model). Note that variational-variance had lower correlation than the L2+L1. Error bars represent standard deviation of n=10 random seeds for each model. We computed the correlation using the predicted and recorded response variances of the test stimuli for each neuron ($r = 0.34, p = 0.002$, Spearman correlation for an exemplary neuron; a). We found that the variational one using response mean as a substitute of variance had a slightly higher mean correlation across neurons compared to the L2+L1 ($p = 0.0368$, two-sided permutation tests on 10 random seeds for 10,000 times; b,c).