# OpenReview forum: "Probabilistic Neural Transfer Function Estimation with Bayesian System Identification"
_ICLR.cc/2024/Conference — ICLR 2024 Conference Withdrawn Submission_

### Official Review · Reviewer_cZAF · 2023-10-31

**Soundness:** 2 fair
**Presentation:** 3 good
**Contribution:** 1 poor
**Rating:** 3
**Confidence:** 4

**Summary:**

The paper uses BetaVAE-style modified variational inference over a neural network to predict calcium imaging responses in two datasets, as well as to compute sensitivity maps for the resultant estimates. The impact of the beta hyperparameter is explored, as does the amount of training data used. The paper claims benefits to prediction quality, uncertainty quantification, and sample efficiency as a result of the variational model.

**Strengths:**

The paper is generally clear in its description of what was done. Further strengths are the use of multiple baselines, and application to multiple datasets.

**Weaknesses:**

My primary concerns fall into two categories: [1] technical concerns about how some of the work was done, and [2] whether there is enough of a new ML contribution provided to be of interest to the ICLR audience.

## Magnitude and significance of the contribution
As far as I can tell, the main contribution here is the application of BetaVAE-style modified variational inference to predict response patterns of calcium imaging data. None of the ideas are particularly new (nor does the paper claim they are), no specialized new insights or techniques were required to get the whole thing to work, and the quality of the results is likewise modest (I wonder if networks with a bit more capacity would perform better). The benefits of some sort of uncertainty quantification for models of calcium imaging may be of use to the neuroscientific community but, even though I love to see neuroscience work at ML conferences, I don't think there's enough here for ICLR.

## Technical concerns
I am puzzled about a few technical choices that I think could have been done better, as well as the core claims.

* Regarding the variational setup -- as far as I can tell, expressions 1-2 describe the conventional ELBO but expressions 3-7 is the BetaVAE ELBO setup (where by turning the problem into constrained minimization they get the beta to show up as a tunable Lagrangian multiplier). For $\beta=1$, the setup is variational inference, but otherwise it's something a bit different (as I understand it), and the paper should make it clearer.
* Regarding the discussion of regularization and sparsity in section 3.1, I am a bit confused. The paper discusses heavy-tailed priors here but reports a mixture of Gaussians prior in section 2.1. Further discussion in this section claims representations become more sparse as $\beta$ increases, which may be true for a sparsifying prior (e.g. spike & slab, Horseshoe, L1), but not for a Gaussian prior as used in the paper. Furthermore, typically sparsity is measured by something like the number of coefficients above zero (or some other threshold), and just because the mean approaches zero and variance shrinks does not imply that the parameters are actually sparse.
* Regarding the predictive performance, I'm surprised the paper does not take advantage of posteriors and take its metrics in expectation over the posterior rather than the point estimate taken from the model (at least, if this is done, it is not stated anywhere). Not doing this seems like a big missed opportunity, and it would be a more principled way to take advantage of quantified uncertainty than measuring the correlation between MEI variance and response CC. Similarly it seems like some sort of proper scoring rule would help measure the quality of the probabilistic predictions.
* Considering there's already an ensemble of models trained, I'm surprised that the paper doesn't take advantage of the ensemble for uncertainty quantification a la Lakshminarayanan et al. NeurIPS 2017.

**Questions:**

Most of my questions are about addressing the technical concerns and confusions above -- please clarify those to the extent possible. In addition, a few minor points:

* I don't think the paper justifies very well its choice of using the L1+L2 model as a testbed to be driven by the other models' MEIs. Why this particular model?
* Can you clarify and explain the claim regarding how matching a Gaussian posterior to a heavy-tailed prior would shrink posterior variance to zero? This does not seem obvious to me. This point is not critical if the paper doesn't actually use a heavy-tailed prior in its empirical work.

---

### Official Review · Reviewer_yDsD · 2023-10-31

**Soundness:** 3 good
**Presentation:** 3 good
**Contribution:** 3 good
**Rating:** 8
**Confidence:** 2

**Summary:**

This paper uses a variational Bayes approach to estimate the parameter uncertainty in the trained convolutional neural network to fit neural responses. The author claims that the model achieves better performance on neural prediction than models without considering uncertainty.

**Strengths:**

The strength of this paper is estimating the uncertainty of the convolutional neural network fitted to neural data. It seems novel to find the parameter uncertainty in a deep network model. The model was also tested by using two neural data sets and compared with other alternatives.

**Weaknesses:**

No significant weakness was found in the paper.

**Questions:**

No.

---

### Official Review · Reviewer_dEtx · 2023-11-09

**Soundness:** 2 fair
**Presentation:** 2 fair
**Contribution:** 2 fair
**Rating:** 3
**Confidence:** 4

**Summary:**

This paper applies a “Bayesian system identification approach” to train models that predict mouse visual neural responses (calcium signals)  to visual stimuli. The authors incorporate weight variability into the trained neural networks as a way to generate a distribution of outputs with interpretable uncertainty. The authors compare this to a baseline deterministic CNN network (which does not give a clear output of uncertainty), and to a network with Monte Carlo dropout. They find that both the weight variability and the network with Monte Carlo dropout explain the neural data better than the baseline network. They also derive Maximum Exciting Inputs for these models.

**Strengths:**

The focus of the paper is topical for the neuroscience and machine learning audience. The idea of introducing some sort of variability in the modeling procedure as a way to approximate distributions is interesting, especially in the context of trying to model stochastic neural data.

**Weaknesses:**

Overall, this paper applies previous work on variational inference to a very specific case of MEIs (and does not introduce new theory, methods, etc). The presented results with the weight distribution also seem not to be much better than the model with dropout. To me, this seems a bit limited in scope for ICLR, especially given the specific weaknesses and questions highlighted below.

W1: The MEIs are not validated with neural data. It seems critical to test that the stimuli successfully drive the neural response, as without this, the generated stimuli may be a “fluke” of the optimization process.

W2: Many of the experimental choices and modeling decisions are not well motivated (some specifics are asked in the Q section below). The authors could generally improve the clarity and presentation of the manuscript.

W3: The statistics in the paper are misleading. There are cases where something is reported as higher than something else, but a fairly large p-value is given (ie the beginning of page 9, with p=0.2114).

W4: The model details description in the appendix has some typos and is not written clearly, which makes it difficult to tell exactly what was done. I’m particularly puzzled by this sentence: “For each CNN model, we tested different numbers (1-5) of nonlinear functions…” as I don’t understand what non-linear functions in the CNNs are being modified.

Minor notes:

M1: Please consider making your plots more colorblind-friendly (I like using https://colororacle.org/ as a checker). As an example, Figure 3b could have different markers in addition to the different colors, making it easier to distinguish the models.

M2: In Figures 3 and 6, it would be more informative if a color bar was put on the MEI and MEI_std, giving the explicit values, instead of having a bolded description in the figure caption that the magnitude of the means is higher than the std.

M3: Much of the wording, notation, and terminology in the paper is a bit non-standard. An example is “CC” being used for “correlation coefficient” (r, as used occasionally in the paper, seems like it would suffice). Aligning the phrasing with commonly used terms in the field would help readability.

**Questions:**

Q1: One of the stated motivations for the work is that it allows for estimation of uncertainty of the MEIs. However, it is not clear to me that you can’t estimate uncertainty from the standard deterministic CNN. For instance, uncertainty can be quantified by initializing the MEI generation from multiple different initializations to better explore the space, or by being estimated on different subsets of the data. I believe that multiple initializations were explored in the previous work in this area, although perhaps uncertainty wasn’t quantified in the same way. Could the authors explain why it is necessary to incorporate some sort of variational inference into the model to capture uncertainty? It would also be useful to quantify how much variability occurs in the MEIs from different initializations in the MEI generation procedure and compare this to the reported MEI_std.

Q2: Why was a distribution on the weights chosen as the way to incorporate a Bayesian approach into the model? Given that dropout seems to have similar effects on the results, it seems like there should at least be a discussion on other ways to add stochasticity into neural networks that might be more neuroscience-inspired (for instance, noise applied to the activations).

Q3: In Figure 3B, does the model with “MC dropout” have dropout turned on during the evaluation process? If so, is the evaluation the same if dropout was turned off for evaluation? Generally, I’m not sure that the “baseline” model is a fair comparison, and something with dropout only used during training (but not evaluation) might be a more clean comparison.

Q4: How is the weight variability handled during the MEI generation? There are known problems with using stochastic responses when doing image synthesis from neural networks, in particular in the adversarial example domain (for instance, Athalye et al. 2018). I suspect that there are similar problems here, where more variability will lead to a landscape that is more difficult to optimize. This raises concerns about the validity of the experiments in Section 3.3, as the results could possibly be explained by difficulties optimizing through the changing weights once the variability gets too large.

Q5: Why was the measure of sparsity in Figure 2 chosen for this setting? It seems a bit ad-hoc, but perhaps a citation would help.